# Novel Mechanisms and Future Opportunities for the Management of Radiation Necrosis in Patients Treated for Brain Metastases in the Era of Immunotherapy

**DOI:** 10.3390/cancers15092432

**Published:** 2023-04-24

**Authors:** Eugene J. Vaios, Sebastian F. Winter, Helen A. Shih, Jorg Dietrich, Katherine B. Peters, Scott R. Floyd, John P. Kirkpatrick, Zachary J. Reitman

**Affiliations:** 1Department of Radiation Oncology, Duke University Medical Center, Durham, NC 27710, USA; 2Division of Neuro-Oncology, Department of Neurology, Massachusetts General Hospital, Boston, MA 02114, USA; 3Department of Radiation Oncology, Massachusetts General Hospital, Boston, MA 02114, USA; 4Department of Neurosurgery, Duke University Medical Center, Durham, NC 27710, USA; 5Department of Pathology, Duke University Medical Center, Durham, NC 27710, USA

**Keywords:** necrosis, brain metastases, stereotactic radiosurgery, immunotherapy, cGAS-STING, artificial intelligence, circulating biomarkers

## Abstract

**Simple Summary:**

As the incidence and survival of patients with brain metastases improve, the burden of treatment-related neurotoxicities will increase for patients and healthcare systems. Radiation necrosis, or injury and inflammation to normal brain tissue, is an increasingly common and deleterious adverse effect of radiation therapy that can contribute to patient morbidity and mortality. We aimed to characterize the biological mechanisms that drive necrosis and the risks associated with multimodal therapy, including immunotherapy. This review additionally provides management guidelines and an overview of novel opportunities for investigation. Awareness of the presentation, risk factors, biological mechanisms, and management options for necrosis are crucial for optimal patient-centered care and discovery.

**Abstract:**

Radiation necrosis, also known as treatment-induced necrosis, has emerged as an important adverse effect following stereotactic radiotherapy (SRS) for brain metastases. The improved survival of patients with brain metastases and increased use of combined systemic therapy and SRS have contributed to a growing incidence of necrosis. The cyclic GMP-AMP (cGAMP) synthase (cGAS) and stimulator of interferon genes (STING) pathway (cGAS-STING) represents a key biological mechanism linking radiation-induced DNA damage to pro-inflammatory effects and innate immunity. By recognizing cytosolic double-stranded DNA, cGAS induces a signaling cascade that results in the upregulation of type 1 interferons and dendritic cell activation. This pathway could play a key role in the pathogenesis of necrosis and provides attractive targets for therapeutic development. Immunotherapy and other novel systemic agents may potentiate activation of cGAS-STING signaling following radiotherapy and increase necrosis risk. Advancements in dosimetric strategies, novel imaging modalities, artificial intelligence, and circulating biomarkers could improve the management of necrosis. This review provides new insights into the pathophysiology of necrosis and synthesizes our current understanding regarding the diagnosis, risk factors, and management options of necrosis while highlighting novel avenues for discovery.

## 1. Introduction

Over 170,000 patients are diagnosed with brain metastases annually in the United States, and up to 40% of cancer patients ultimately develop intracranial metastases [1,2]. Patients with lung, breast, melanoma, renal, and colorectal cancer are at particular risk for developing brain metastases. Over the last two decades, novel systemic agents have significantly improved patient outcomes, particularly with the discovery of targeted therapies against driver mutations and immune-checkpoint inhibitors [3]. Even so, radiation therapy, with or without surgery, remains the standard of care for the local management of intracranial disease [4]. Due to improvements in patient survival, oncologists are now increasingly tasked with managing the long-term radiation-associated toxicities of CNS-directed therapies. With the transition from whole-brain radiotherapy (WBRT) to stereotactic radiosurgery (SRS) for patients with multiple brain metastases, the nature of these CNS-toxicities has also evolved.

SRS is a specialized radiation technique that delivers a single or few fractions of high-dose radiation to a small target volume. Technological advances allow SRS planning systems to deliver radiation with submillimeter precision to multiple targets while avoiding critical structures or organs at risk (OARs) such as the optic nerves, optic chiasm, and brainstem [5]. However, the brain abutting the target receives a modest ring of high-dose radiation. SRS most commonly employs photons as the radiation type, which is most commonly delivered by either a linear accelerator (LINAC; multiple vendors) or a Gamma Knife (Elekta; Stockholm, Sweden) treatment system. Multiple other commercial systems, including Cyber Knife (Accuray; Sunnyvale, CA, USA), exist that employ the same concept of converging multiple low megavoltage radiation beams. Frameless image-guided stereotactic systems employ mask-based fixation approaches, which are increasingly replacing invasive cranial fixation. Radiation doses are based on tumor dimension and range from 15–24 Gy when delivered in a single fraction [6]. Hypofractionated approaches are often considered a safer treatment for larger lesions or resection cavities and are delivered in doses of 25–35 Gy in three to five fractions [7,8]. Local control rates for brain metastases treated with SRS are >90%, though rates of distant intracranial relapse are increased compared to WBRT [9]. Multiple clinical trials have demonstrated that photon-based SRS treatments are an appropriate option for patients with multiple brain metastases and are associated with improved neurocognitive outcomes compared to WBRT due to the sparing of normal brain tissue [10,11,12,13]. Despite the theoretical dosimetric advantages of protons, their physical limitations when treating small target lesions and their inferior cost-effectiveness in adult patients have minimized their impact on brain metastasis patients [14].

Brain metastases are increasingly co-managed with SRS and systemic therapy to maximize local and extracranial disease control. This shift in practice necessitates a careful evaluation of the additive treatment-related toxicities associated with combined modality approaches. Among the most harmful adverse effects of SRS is radiation necrosis. In this review, we discuss emerging insights into the biological mechanisms linking radiation-induced DNA damage and necrosis. Additionally, we critically review the latest data on toxicity when SRS is delivered concurrently with novel systemic agents and highlight future opportunities involving novel diagnostic imaging platforms, artificial intelligence, and circulating biomarkers.

## 2. Pathophysiology

Radiation necrosis, increasingly known as treatment-induced necrosis, refers to a delayed and typically progressive and irreversible form of radiation-induced inflammation or injury to normal brain parenchyma [15]. The exact pathophysiology of radiation necrosis is multifactorial and incompletely understood, with glial and endothelial cell injury implicated as key mediators. Recent insights into signaling pathways linking radiation-induced DNA damage and immune responses have shed light on processes that could be equally critical to necrosis development, thus providing novel opportunities for discovery.

### 2.1. cGAS-STING Pathway

Recent insights suggest that activation of the cyclic GMP-AMP (cGAMP) synthase (cGAS) and stimulator of interferon genes (STING) pathway (cGAS-STING) could explain how radiation-induced DNA damage triggers pro-inflammatory cascades and injury to normal brain tissue. Ionizing radiation induces lethal double-strand DNA (dsDNA) breaks either directly or indirectly through the production of reactive oxygen species (ROS), triggering a cascade of DNA damage repair (DDR) pathways. Without adequate repair, irreparably damaged normal and cancerous cells undergo mitotic catastrophe, apoptosis, autophagy, or enter cellular senescence [16]. Damaged dsDNA fragments are sensed by the cGAS-STING pathway using cytosolic nucleic sensors. The mechanism by which these cytosolic sensors promote innate and adaptive immune responses has recently received considerable attention [17,18,19,20,21,22,23,24,25,26,27].

Dendritic cell (DC) activation and type 1 interferon (IFN) signaling are thought to be tightly linked to cGAS-STING activation [24,27,28,29,30,31,32,33]. Both are essential to innate immunity and provide a critical link between innate and adaptive anti-tumor immune responses [34,35,36]. CD8α+ DCs, a subset of conventional DCs type 1, are crucial for the cross-presentation of extracellular tumor antigens on MHC I molecules to cytotoxic CD8+ T cells, which subsequently recognize target cells and trigger apoptosis through the release of granzyme B and perforins or through activation of death receptor signaling (i.e., Fas/FasL) [37,38]. Type I IFNs, including IFNα and IFNβ, activate antiproliferative cellular processes and are critical for DC functioning [39], including DC maturation, costimulatory molecule expression, and migration through lymphatics [40,41]. In preclinical models of immunogenic tumors, type I IFNs were essential for regulating the capacity of CD8α+ DCs to prime CD8+ T-cells and facilitate anti-tumor responses [40,42,43,44].

dsDNA molecules in the cytosol of normal and malignant cells act as immunostimulatory damage-associated molecular patterns (DAMPs) and are essential for the activation of the cGAS-STING pathway. DNA is thought to accumulate in the cytosol following radiation therapy via several possible mechanisms. Ionizing radiation leads to the formation of micronuclei containing dsDNA, which break down in the cytoplasm and then release their nucleic acids [17,18]. Harding et al. showed that cell cycle progression through mitosis following dsDNA breaks induced by radiation leads to micronuclei formation, an event that precedes the activation of inflammatory signaling [18]. Other sources of dsDNA include chromatin fragments released from the nucleus due to nuclear lamin B1 degradation or cytosolic leakage of oxidized mitochondrial DNA [21,22,23,33,45,46,47,48,49,50]. Alternatively, it has been suggested that exosomes containing dsDNA are released by damaged cells and ingested by DCs and neighboring cells [51].

In the presence of dsDNA, cytosolic cGAS undergoes conformational changes that expose its catalytic pocket. This allows for the conversation of GTP and ATP molecules into the messenger molecule, 2′,3′-cGAMP [17,18,24,52], which then binds STING in the endoplasmic-reticulum [28,30]. 2′,3′-cGAMP can translocate through gap junctions into neighboring cells, including DCs, to function as a paracrine signaling molecule [53]. In the presence of cGAMP, STING oligomerizes with TANK-binding kinase 1 (TBK1), leading to the phosphorylation of STING and recruitment of interferon regulatory factor 3 (IRF3) [54,55,56,57]. Phosphorylation of IRF3 by TBK1 results in dimerization and translocation of IRF3 to the nucleus, where it acts as a transcriptional factor for the expression of type I IFNs and inflammatory responses [25,33,58,59]. Activation of innate immune responses ultimately leads to the selection of tumor-antigen-specific CD8+ T-cells and the accumulation of CD4+ T-cells, macrophages, and DCs in the tumor microenvironment (Figure 1) [60,61,62].

While the cGAS-STING pathway is crucial for coordinating appropriate immune responses, overactivation may contribute to adverse effects, including radiation necrosis. Under normal conditions, lysosomal degradation, ubiquitination, and inhibitory phosphorylation of STING dampen cGAS-STING activation [63,64,65]. However, when unchecked, this pathway has been implicated in autoimmune disorders such as Aicardi–Goutières syndrome [66,67]. In patients with brain metastases treated with SRS, overzealous activation of cGAS-STING may similarly effectuate the development of a pro-inflammatory tumor microenvironment and damage normal tissues. This pathway may explain the elevated levels of type 1 IFNs, IL-1, IL-6, IL-8, VEGF, EGFR, TNFα, ROS, and cytokines in necrotic brain tissue following radiation [68]. Biopsy specimens contain inflammatory components and evidence of injury, including histiocytic infiltrates, reactive gliosis, hemorrhage, thrombosis, vascular abnormalities, fibrinous exudates, and signs of parenchymal necrosis [69]. These observations suggest a complex process mediated by both glial and endothelial cell injury, resulting in hypoxia-induced VEGF overexpression and activation of immune cell infiltration (Figure 2).

Novel therapeutics targeting the cGAS-STING pathway are the subject of active investigation for patients with brain tumors. According to a pan-cancer analysis, this pathway is frequently disrupted by hypermethylation or loss-of-function mutations at cGAS or STING promoters [70]. Preclinical models suggest that DNA methyltransferase inhibitors and STING agonists may rescue the cGAS-STING pathway, allowing remodeling of the tumor microenvironment, improved anti-tumor immune responses, and more durable clinical outcomes [71,72,73]. However, the potential toxicity of these therapeutics in patients, particularly in the setting of radiotherapy, is unknown and must be evaluated with caution.

Animal models for radiation necrosis have been proposed to identify imaging biomarkers [74,75,76]. Future research should leverage these preclinical models to elucidate the importance of cGAS-STING and other implicated pathways in radiation necrosis development. How the tumor microenvironment, particularly for immunogenic tumors such as melanoma, interacts with these mechanisms should also be investigated [77]. While clinical studies suggest an elevated risk of necrosis with melanoma brain metastases, the biological processes underpinning potential differences in necrosis incidence between tumor types are unknown and require representative model organisms for proper interrogation [78,79]. This represents an important avenue for future investigation, particularly as immune-modulating therapies continue to enter the treatment landscape.

### 2.2. Temporal Patterns

Necrosis is generally a late, delayed treatment-related complication occurring at a median of twelve months following irradiation, though early and very late events are reported in the literature [15]. Immunotherapy may potentiate necrosis risk for patients with metastatic brain tumors. Radiation is known to augment innate and adaptive immunity, potentially via the cGAS-STING pathway. SRS acts as a primer, upregulating MHC I expression on tumor cells, inducing immunologic death, increasing T-cell infiltration, and stimulating type I IFN production [43,80,81,82,83]. These effects may be enhanced in the setting of an immune checkpoint inhibitor, thus potentiating both treatment efficacy and risk for unwanted injury of normal brain tissue, as observed by several groups [78,79,84].

## 3. Diagnostic Evaluation

Radiation necrosis poses a major diagnostic challenge, given its similarity to recurrent disease on conventional neuroimaging. Radiographically, necrosis typically occurs within the high-dose radiation treatment volume and appears as a focal area of increasing peripheral enhancement. Necrotic lesions may exhibit a “soap bubble”-like pattern of enhancement on T1-weighted post-gadolinium MRI sequences (Figure 3). This can be accompanied by surrounding T2-weighted hyperintensity due to vasogenic edema. Reliable differentiation of necrosis from disease recurrence on structural MRI alone is seldom possible. Some propose calculating a “lesion quotient” by determining the ratio of T2 nodularity and contrast enhancement on T1 sequences. Ratios less than 0.3 may correspond with necrosis [85,86]. Brain metastases in the corpus callosum and periventricular white matter may be at increased risk for necrosis due to the greater susceptibility of these sites to microvascular injury; however, the association between brain location and necrosis risk remains a subject of ongoing investigation [15]. As imaging biomarkers from conventional MRI are frequently misleading [15], advanced imaging modalities are increasingly integrated into the diagnostic process. However, surgical biopsy remains the gold standard to distinguish suspected necrosis from tumor progression. Occasionally, biopsies may contain mixed elements of both necrosis and foci of tumor cells or scattered atypical cells, further confounding pathologic confirmation [15]. The clinical implication of these “mixed findings” for tumor control is unknown.

## 4. Clinical Significance

Nearly half of patients with necrosis develop generalized or focal neurologic symptoms, termed symptomatic necrosis [87,88,89]. These rates vary based on the intracranial location of lesions and often are greater with larger irradiated volumes, higher biologically effective doses (BED), and immunotherapy. Neurologic deficits occur due to mass effect secondary to vasogenic edema or disruption of normal brain parenchyma in eloquent brain regions. Presenting symptoms may include global cognitive changes, altered mental status, headaches, focal neurologic deficits, and seizures [15,90].

As treatment of radiation necrosis differs sharply from that of recurrent disease, accurate and timely diagnosis of either entity is paramount. Whereas tumor progression necessitates prompt antineoplastic therapy in the form of reirradiation, surgery, or a change in systemic therapy, symptomatic necrosis is often managed with corticosteroids, pentoxifylline, vitamin E, bevacizumab, surgical resection, or laser interstitial thermal therapy (LITT) [15]. Given the differences in management, accurate diagnosis is critical in order to maximize oncologic outcomes and avoid unnecessary treatment-related morbidity. The misdiagnosis of necrosis as progressive disease may result in unnecessary delivery of antineoplastic treatment, which otherwise could have been reserved as salvage therapy. Vice versa, treatment options for necrosis can produce adverse effects that potentially undermine the efficacy of systemic therapy.

The importance of accurate diagnosis and management is underscored by several studies. In a cohort of 640 advanced non-small cell lung cancer (NSCLC) patients, those receiving ≥10 mg of prednisone equivalent within 30 days of starting anti-PDL-1 therapy had worse progression-free survival and overall survival [91]. This negative association was maintained even after controlling for smoking history, performance status, and the presence of brain metastases. Kotecha et al. similarly reported worse survival when patients treated with SRS and concurrent immune-checkpoint blockade were treated with steroids (5.1 vs. 10.2 months, *p* = 0.002) [92]. In both studies, the anti-inflammatory and immunosuppressive effects of corticosteroids may have counteracted the efficacy of immune-checkpoint inhibitors, contributing to worse clinical outcomes. With the increased use of immune-activating agents, efforts to minimize radiation necrosis are essential. Additionally, accurate non-invasive strategies for diagnosing and selecting appropriate therapies are needed.

Ultimately, clinicians are tasked with the delicate balance of optimizing tumor control while minimizing adverse effects. Preventing radiation necrosis should not come at the expense of durable cancer control, particularly when there are adequate treatments that can address necrosis. However, as the incidence and survival of brain metastasis patients rise, radiation necrosis will remain a growing clinical challenge for radiation oncologists, neuro-oncologists, medical oncologists, and neurosurgeons. An appreciation of the risk factors, management, and future opportunities for innovation is critical, especially in the era of novel systemic agents.

## 5. Risk Factors for Necrosis

### 5.1. Radiation Therapy Alone

SRS is well-tolerated and historical data suggest low necrosis rates following treatment of intact and resected brain metastases with radiation alone (Table A1) [93,94,95]. More modern series suggest that the rate of any grade necrosis is less than 10% with SRS alone for intact metastases, though up to 54% of necrotic lesions can be symptomatic [87]. Across studies, tumor size has emerged as an important predictor for necrosis. For larger lesions, fractionated SRS in up to five fractions (fSRS), also known as hypofractionated stereotactic radiation therapy (HF-SRT) or hypofractionated SRS (HF-SRS), may lower the incidence of necrosis. Minniti et al. found that fSRS (27 Gy/3 fx) was associated with reduced radiographic necrosis compared to SRS (8% vs. 20%, *p* = 0.004). Additionally, necrotic lesions treated with fSRS were less likely to be symptomatic (41.9% vs. 36.4%, *p* = 0.04). Tumor volume, V12Gy (SRS), and V18Gy (fSRS) predicted necrosis risk on multivariate analysis [88]. This is consistent with other smaller studies that report the importance of V10Gy and V12Gy as predictors of necrosis risk following SRS [96]. In the postoperative setting, necrosis rates are comparable to those seen with intact lesions and range from 9% to 23% with radiation alone (Table A2) [97,98]. Resection cavities may similarly benefit from fSRS. In a series of 160 patients treated with 24–30 Gy in three to five fractions, only 8.9% of resection cavities developed necrosis, of which two (15.4%) cases were symptomatic [99]. Reirradiation in this study was predictive of necrosis. Eitz et al. also reported similar rates in a larger series of 558 patients treated with fSRS to a median dose of 30 Gy (range: 18–35) [100]. Only 6.3% of necrotic lesions were symptomatic.

Preoperative SRS is an active area of investigation and may further reduce toxicity. Advantages of preoperative radiotherapy include better target volume delineation and possibly improved tumor control by reducing intraoperative seeding and leptomeningeal disease. Emory University and the Levine Cancer Institute reported outcomes from 180 patients undergoing preoperative (36.7%) versus postoperative (63.3%) SRS [101]. Postoperative SRS was associated with elevated symptomatic necrosis at two years (16.4% vs. 4.9%, *p* = 0.010). This corroborates data from Prabhu et al., who reported a necrosis rate with postoperative, preoperative, and non-operative SRS at 1 year of 22.6%, 5%, and 12.3%, respectively (*p* < 0.001) [101]. The randomized phase III NRG trial BN012 (NCT05438212) opened in June 2022 and will compare preoperative versus postoperative SRS outcomes for resected brain metastases.

### 5.2. Reirradiation

The etiology for local recurrence after SRS is likely multifactorial and driven by treatment-related factors and tumor biology, including the presence of cancer stem cells [102,103]. Select patients with recurrent brain metastases may be treated with reirradiation with the risk of necrosis with retreatment as the salient consideration. The recovery rate of normal brain tissue following radiation treatment is ill-defined and requires consideration of total radiation dose, fractionation, irradiated volume, location in the brain, patient co-morbidities, and cross-interaction with other treatments. The strongest data on reirradiation toxicity comes from RTOG 9005, a multi-institutional trial of 156 patients with radiographically recurrent primary (36%) and metastatic (64%) cerebral or cerebellar tumors [6]. Maximum tolerable doses were 24 Gy, 18 Gy, and 15 Gy for tumors < 20 mm, 21–30 mm, and 31–40 mm in maximum diameter. 16 (10%) patients developed necrosis, 15 (94%) of whom necessitated surgical intervention. Tumor diameter and radiation dose were both predictive of necrosis. More modern series document higher rates of any grade necrosis following reirradiation, with rates ranging from 13.4% to 38.4% (Table A3) [104,105,106]. Additionally, radiation dose, lesion size, and prior WBRT have emerged as important risk factors for necrosis in this setting [104,105,106]. The elevated necrosis rates on retrospective studies, combined with the heterogeneous tumor histologies, lack of biopsy-confirmed recurrent tumor, and high rates of symptomatic necrosis seen on RTOG 9005, warrant careful counseling and surveillance for patients undergoing reirradiation. Limitations of these studies include the infrequent rate of biopsy-confirmed necrosis and the inconsistent reporting of systemic therapy, which may have contributed to the elevated necrosis rates in recent studies. Studies documenting long-term outcomes with reirradiation in the setting of novel systemic agents are necessary.

### 5.3. Immunotherapy

Immune-checkpoint inhibitors (ICI) are increasingly used as single agents (e.g., anti-CTLA-4, PD-1, and PDL-1) or as combined/dual agent therapy alongside SRS for the treatment of NSCLC, melanoma, renal cell carcinoma (RCC), and other histologies of brain metastases [107,108,109,110,111]. SRS delivered concurrently with immunotherapy may provide a synergistic benefit, but these approaches may also increase toxicity (Table 1) [112]. The best data comes from a prospective phase II trial evaluating the efficacy of pembrolizumab in 23 melanoma patients with at least one asymptomatic untreated metastasis measuring 5 mm to 20 mm [84]. Prior resection or radiation treatment to other lesions was permitted. Despite only 52% of patients receiving prior SRS, seven (30.4%) patients developed necrosis, five of which were biopsy-proven. The mean time from SRS to necrosis was 19.4 months. This elevated incidence was attributed to pembrolizumab exposure and improved patient survival, which allowed longer follow-up.

Similarly, retrospective studies report necrosis rates up to 37.5% with single-agent ICI and SRS. One group of investigators reported that ICI (HR: 2.56; 95%CI: 1.35–4.86) and melanoma histology (HR: 4.02; 95%CI: 1.17–13.82) significantly increased radiation necrosis risk in a cohort of 480 patients. No patients received dual ICI, and the timing of SRS relative to ICI was not reported [78]. In another series of 180 patients, any grade necrosis and symptomatic necrosis rates were 21.7% and 46.2%, respectively, across the entire cohort [113]. 12 (37.5%) patients treated with single-agent ICI developed radiographic necrosis and single ICI was associated with necrosis risk (OR: 2.71; 95%CI: 0.94–7.76, *p* = 0.06). In another series of 80 patients, Minniti et al. reported that 35% of patients developed necrosis with either ipilimumab or nivolumab plus SRS or fSRS delivered within one week [114]. While these data suggest a potential increase in toxicity with combined SRS and immunotherapy, other studies did not detect significant associations [115,116,117]. Discrepancies are likely driven by sample size and inadequate power. Nonetheless, most series consistently observe an earlier onset of necrosis and higher frequencies of symptomatic necrosis with the addition of single-agent ICI.

The optimal temporal sequencing of immunotherapy and SRS is unknown. Concurrent ICI and SRS are frequently defined as treatment delivered within four weeks of either modality [7]. Goldberg et al. reported outcomes from a prospective series of 42 NSCLC patients treated with pembrolizumab [118]. Twenty-one patients had received prior radiation, of whom seven received SRS within 6 months of immunotherapy. All three cases of necrosis in the trial occurred in those patients treated with pembrolizumab within six months of SRS, suggesting a potential interaction between the timing of immunotherapy and radiation therapy. However, a recent retrospective, multi-institutional series of 657 SRS patients observed comparable rates of necrosis with concurrent and delayed ICI. Importantly, the study lacked a control group treated with SRS alone, excluded patients treated with fSRS, and included only a small proportion of patients treated with dual ICI (15.8%) [119]. In contrast, a single institution series of 206 NSCLC and melanoma patients found that concurrent ICI and SRS within 4 weeks were associated with increased necrosis risk compared to SRS alone (HR: 6.47; 95%CI: 3.60–11.62) [79]. 35% of patients received dual ICI, and rates of symptomatic necrosis were significantly elevated in this cohort (dual: 36.4%, single: 17%, and none: 13.7%, respectively, *p* < 0.001). Melanoma (HR: 2.41; 95%CI: 1.32–4.63) and hypofractionation (HR: 0.27; 95%CI: 0.13–0.54) were also predictive of necrosis. These findings align with those observed by several groups [78,84,114]. Discrepancies between studies underscore the need for large prospective trials to evaluate the toxicity and efficacy of concurrent treatment and use of dual ICI in brain metastasis patients receiving SRS.

### 5.4. Targeted Therapy

Radiation therapy combined with targeted systemic therapy for brain metastases appears well tolerated, based on a limited set of retrospective studies evaluating patients with NSCLC, melanoma, and breast cancer. Most studies report necrosis rates of 1% to 8% in the setting of targeted agents, including vemurafenib and lapatinib (Table 2) [120,121,122,123]. However, Park et al. reported a necrosis rate of 61.9% in a small cohort of 46 HER2+ breast cancer patients [124]. Targeted therapy was delivered within four weeks of radiation in 58.7% of patients, and the use of multiple HER2-directed therapies predicted necrosis. Patients with radiation necrosis were more likely to have received multiple HER2 agents during radiotherapy (35.7% vs. 5.6%, *p* = 0.047). Standard practice is often to hold targeted agents, especially tyrosine kinase inhibitors, for several half-lives before and after SRS to avoid toxicities that may be associated with the concurrent administration of both treatments. Though the current literature suggests an excellent safety profile of targeted therapy and SRS, findings by Park et al. highlight the need for careful continued evaluation.

### 5.5. Chemotherapy

Chemotherapy has previously been associated with increased toxicity in patients receiving radiation therapy. Radiation recall, an acute inflammatory reaction limited to previously irradiated areas, is an uncommon event that can occur days to weeks following exposure to drugs such as anthracyclines, antimetabolites, and taxes [125]. Even patients with remote histories of radiation therapy are at risk for radiation recall, which commonly manifests as acute dermatitis. Though radiation recall is rare and the underlying mechanism appears to be distinct from radiosensitization, it raises the possibility that chemotherapy also increases the risk for brain radiation necrosis.

Radiation practice patterns in the setting of chemotherapy are based on decades of prior experience that predate modern radiation approaches. The analysis of contemporary series suggests mixed observations regarding the association between chemotherapy and radiation necrosis (Table 3). Cagney et al. reported a significantly elevated 1-year rate of necrosis when SRS was delivered alongside pemetrexed-based chemotherapy (24.1% vs. 9.8%; HR: 2.70; 95%CI: 1.09–6.70, *p* = 0.03) [126]. This is consistent with another group in which necrosis in 24% of lesions in the setting of chemotherapy was reported [127]. Another series of 435 patients found that capecitabine/5FU delivered within 4 weeks of SRS was associated with significantly increased necrosis risk (HR: 3.61; 95%CI: 2.90–4.51) [128]. Prior radiation exposure, target volume, and taxane therapy were also predictive. Studies that did not detect an elevated risk of necrosis with chemotherapy nonetheless also observed a higher overall incidence, including symptomatic necrosis, compared to historical series of SRS alone [89,129,130]. Further investigation of necrosis outcomes following chemotherapy and SRS is necessary. Clinicians should counsel patients appropriately when offering SRS in this setting.

### 5.6. Proton Therapy

The added value of proton SRS (as opposed to more conventional photon-based SRS) for the treatment of most brain metastases remains questionable and likely limited to select patients [14]. In a cohort of 370 patients, Atkins et al. demonstrated reasonable local control and minimal acute toxicity with protons [131]. However, data from the primary brain tumor literature suggests that protons carry an increased risk for radiation necrosis compared to photons, particularly at the distal beam regions of high linear energy transfer (LET) due to the greater relative biological effectiveness (RBE) of protons (RBE of 1.1) [132,133,134]. In practice, the inherent characteristics of protons limit their application to brain metastasis patients. Unlike cost-effectiveness analyses in pediatric populations, analyses in patients with brain metastases who are almost entirely adult and elderly do not currently favor protons [135]. Accurate proton dosimetry is limited by 2 mm to 3 mm and 2% to 3% range uncertainty, which is a clinically meaningful distance in SRS. However, patients with large, irregular tumors or resection cavities adjacent to critical structures may benefit from improved conformity and dose escalation with protons [136]. Young, fit patients with brain oligometastases may also benefit, though this population is likely small. Despite the limited application of protons for brain metastases, awareness of the underlying dosimetric advantages and disadvantages is appropriate.

## 6. Management Options

No standard of care exists for the management of radiation necrosis, though corticosteroids are the first and most common intervention used to reduce symptoms associated with vasogenic edema and mass effect. However, steroids are limited by their adverse side effect profile and may interfere with the efficacy of immunotherapy. Other non-invasive treatments include hyperbaric oxygen therapy, pentoxifylline, vitamin E, and nerve growth factor [137,138,139,140]. Bevacizumab, a VEGF-A monoclonal antibody, is a non-invasive alternative that has shown promise in reversing radiographic changes and neurologic deficits [141,142,143,144,145,146,147]. The efficacy of bevacizumab stems from the proposed mechanism by which radiation therapy stimulates angiogenesis, vascular permeability, and brain edema through the induction of hypoxia-inducible factors (HIF1α) and VEGF. Bevacizumab-mediated binding and downregulation of VEGF are thought to treat necrosis by interrupting pathways downstream of tissue hypoxia [141].

Surgical intervention may be necessary for acutely symptomatic cases and can be preferred over more conservative medical management to accelerate clinical recovery from more mild symptoms. In a single institution series of 46 patients, resection significantly reduced steroid dependency from 54% preoperatively to 15% postoperatively at 12 months (*p* = 0.001) [148]. LITT is a minimally invasive alternative option for surgically inaccessible lesions and when a craniotomy is contraindicated [149]. LITT systems treat necrosis using hyperthermia [150] and are at least equally effective to bevacizumab or open resection [151,152,153,154]. This emerging approach is the subject of the multi-institutional REMASTer trial, which will compare LITT to steroids for the treatment of biopsy-confirmed necrosis following SRS (NCT05124912).

## 7. Future Directions

### 7.1. Dosimetric Strategies

Treatment strategies that limit toxicity without compromising tumor control are ideal. Several studies comparing conventional SRS with fSRS report lower rates of necrosis and excellent oncologic outcomes. Using the linear-quadratic model for the estimation of the dose-effect relationship, Minniti et al. delivered an equivalent SRS dose of 22 Gy (assuming an α/β of 12 Gy for brain metastases (BED_12_)) using fSRS doses of 27 Gy in 3 fractions [88]. Doses were prescribed to the 80% to 90% isodose line with a minimum 95% target coverage, and fSRS was associated with reduced necrosis. In a Korean series of 105 patients, Chon et al. observed 1-year rates of 39.8% and 0% with SRS (median: 20 Gy to 40% isodose line, range: 18–22) and fSRS (median: 35 Gy to 80% isodose line, range: 27–41) delivered in three or five daily fractions, respectively [155]. This is consistent with systematic reviews and meta-analyses reporting improved oncologic outcomes and reduced toxicity with fSRS [112,156]. Based on these findings, hypofractionation represents a compelling strategy to balance neurotoxicity with oncologic control, particularly for metastases ≥3 cm or lesions located in critical areas.

For both single and multi-fraction SRS, dosimetric guidelines exist to guide treatment planning [8]. A V12 Gy (normal brain plus tumor) of 5 cm^3^, 10 cm^3^, or >15 cm^3^ is associated with symptomatic necrosis rates of 10%, 15%, and 20% for SRS plans. QUANTEC and UK SABR also recommend limiting V12 to less than 10 cm^3^ [157]. For fSRS, a V20 (3-fractions) or V24 (5-fractions) <20 cm^3^ carries a <10% risk of any necrosis or edema. The small PTV margin (1 mm to 2 mm) used in SRS plans allows providers to meet these dose constraints without compromising clinical outcomes [158]. Interestingly, some data suggest that LINAC-based treatments may further reduce radiation necrosis incidence compared to Gamma Knife treatments, possibly due to the increased inhomogeneity and internal “hot spots” with Gamma Knife [159]. Ultimately, appropriate image guidance and patient setup coupled with a robust quality assurance program are crucial to optimize outcomes. Future research should consider refining predictive models and dosimetric guidelines in select high-risk patient populations, particularly those receiving reirradiation or concurrent immunotherapy.

### 7.2. Novel Imaging Techniques

Several novel imaging approaches have been pursued to improve conventional MRI techniques. MR diffusion and perfusion-weighted scans are one strategy to discern recurrent tumors from radiation necrosis by detecting the increased vasculature associated with neoplasms [160,161]. However, inter-provider variability in interpretation remains a limiting factor with this approach [162,163]. Delayed contrast MRI, which exploits the preferential blood supply seen in recurrent tumors, is the subject of a clinical trial currently open to accrual (NCT04246879).

Radiotracers have emerged as another promising strategy, as tumor cells exhibit increased uptake, possibly due to elevated expression of amino acid transporters [164]. This has led to the evaluation of radiolabeled amino acids to selectively identify tumor cells. In a cohort of 42 patients with 50 brain metastases, investigators reported significant improvement in diagnostic accuracy using F-DOPA scans (sensitivity 90%, specificity 92.3%) [165]. Over time, recurrent tumors demonstrated an increase in F-DOPA uptake relative to necrotic lesions [166]. A prospective study of ^11^C-methionine positron emission tomography (MET-PET) in 32 patients reported a sensitivity and specificity of 82% and 75% [167]. An open-label single-arm phase II trial (NCT04410367) that recently completed accrual aims to establish image interpretation criteria for 18F-fluciclovine PET studies in patients scheduled for craniotomy to resect a treated brain metastasis that is equivocal on MRI for recurrence [168]. This synthetic amino acid has preferential uptake in tumors and is now the subject of an open phase III trial (NCT04410133) in brain metastasis patients. Finally, MR spectroscopy has also been used to evaluate intralesional metabolite concentrations, which are hypothesized to differ between recurrent tumors and normal tissue. Increased levels of choline creatinine, choline-N-acetyl aspartate, and lipid-lactate levels may be associated with radiation necrosis [169].

Though promising, these imaging modalities remain investigational. Additionally, the variability in image acquisition protocols between institutions presents a barrier to the interpretation of single-institution trials and the integration of findings into practice guidelines. The Deauville criteria, widely used for staging and assessment of treatment response in lymphoma patients, provide a template that clinicians can look to as a strategy to overcome these limitations. Standardizing imaging criteria and demonstrating the clinical utility of these novel techniques should remain a focus of future research.

### 7.3. Artificial Intelligence

Computational image analysis using radiomics and classic machine learning may further improve the accuracy of current diagnostic approaches [170,171,172,173]. These methods leverage computational image-based biomarkers such as pixel intensity, shape, size, or volume to inform supervised machine learning algorithms. However, variability in acquisition parameters, software, and imaging equipment, as well as imbalanced datasets, limit the accuracy of radiomics-based models. Deep learning represents a powerful tool for medical image analysis that incorporates advances in computational hardware, algorithms, and big data analysis [174,175]. In contrast to manually selected radiomics features, deep learning selects abstract and high-throughput hierarchical information using heuristic modeling [174]. Deep learning is increasingly combined with radiomics approaches, with their well-characterized mathematical expressions, to improve algorithm performance and interpretability. Recent applications of radiomics-informed deep learning models in the context of diagnostic platforms for recurrent brain tumors are encouraging.

Recently, investigators developed a radiomics-informed deep learning model using a cohort of 51 brain metastasis patients with biopsy-confirmed recurrence (n = 14) or radiation necrosis (n = 37) after SRS. Their model significantly outperformed historical radiomics approaches, with a sensitivity and specificity of 65% and 64%, with an AUC of 0.69 [176]. Using a different radiomic signature combined with machine learning, Chen et al. achieved a sensitivity, specificity, and AUC of 52%, 90%, and 0.71 (95%CI: 0.51–0.91) using a cohort of 135 brain metastases (n = 40 biopsy confirmed) [172]. Another group of investigators reported similar outcomes using a radiomics approach that considered shape, intensity, neighborhood intensity difference (NID), grey-level co-occurrence matrix (GLCM), and grey-level run-length matrix (GLRLM) [177]. A final model using post-SRS brain metastasis surface area and texture (GLCM 3D Homogeneity) and pre-SRS brain metastasis roundness predicted necrosis with an AUC of 0.71. However, the lack of biopsy confirmation in this study limits interpretability.

Advancements in image analysis and deep learning, combined with radiomics approaches, will continue to improve existing algorithms. To augment clinical utility and reliability, future research should limit training sets to biopsy-confirmed cases and carefully document adverse events in the context of prospective trials. Computational imaging-based approaches should be integrated with biologic insights, including circulating biomarkers, to enhance model performance.

### 7.4. Circulating Biomarkers

Peripheral biomarkers for radiation necrosis have yet to be evaluated in patients with brain metastases. Data on circulating biomarkers are generally limited to primary brain tumors. Radiation necrosis entails a complex sequence of events that results in the upregulation of pro-inflammatory mediators, including type 1 IFNs, TNFα, interleukins, ICAM-1, and VEGF [68]. Markers of blood-brain barrier damage and neuronal injury include S100 calcium-binding protein B and neuron-specific enolase. These markers could potentially be detected in peripheral blood. In glioblastoma, radiation damage to normal brain tissue may induce differential expression patterns among myeloid-derived suppressor cells, thus providing a peripheral blood signature for radiation necrosis [178]. Other studies suggest that extracellular microvesicles may accurately select for glioblastoma recurrence following chemoradiation [179].

Discerning whether circulating biomarkers originate from extracranial versus intracranial disease is an inherent challenge for patients with brain metastases. Methylation profiling of cell-free DNA may overcome this obstacle. All cell lineages possess unique DNA methylation patterns to appropriately regulate cell-specific gene expression. These differentially methylated regions, detectable in nucleic acids released into the blood, correspond to the cell of origin and even the cell state. DNA methylation patterns have already been used to predict treatment response and guide therapy selection for colon, liver, breast, and prostate cancer patients [180,181,182,183,184,185]. Epigenetic signatures may similarly predict outcomes in brain tumor patients [186]. Sabedot et al. demonstrated that glioma patients had elevated serum cell-free DNA levels and distinct genome-wide cell-free DNA methylation patterns [186]. The authors developed a glioma epigenetic liquid biopsy (GeLB) score and found that glioma serum methylomes clustered with primary glioma tissue methylation patterns, suggesting that these patterns can give insight into the cell of origin. Intriguingly, GeLB scores declined significantly in three patients with biopsy-proven pseudoprogression. These findings have implications for patients with brain metastases and could enable clinicians to develop a brain metastasis-specific marker for radiation necrosis.

Finally, deconvolution approaches using peripheral blood methylation data can offer investigators insight into the activation state of peripheral immune cell populations. Inference of CD4+ T-cell, CD8+ T-cell, B-cell, natural killer, monocyte, and granulocyte proportions is achievable with cell-free DNA methylation data. In a study of 72 glioma patients, Wiencke et al. reported elevated neutrophil-to-lymphocyte ratios and inferred natural killer cell activation status using differentially methylated regions [187]. Sabedot also observed differences in the composition of immune cell-specific methylation signatures in patients with different brain tumor histologies using deconvolution methods [186]. The upregulation of pro-inflammatory pathways and recruitment of immune cells seen with radiation necrosis may manifest as alterations in peripheral blood immune-cell subpopulations, thus providing a potential novel signature for necrosis. These approaches are the subject of ongoing prospective trials (NCT05480644 and NCT05695976) and necessitate further evaluation in the context of a multi-institutional collaboration.

## 8. Conclusions

Radiation necrosis, increasingly referred to as treatment-induced necrosis, represents a growing neuro-oncologic challenge as survival improves for brain metastasis patients. Recent mechanistic insights linking radiation-induced DNA damage via the cGAS-STING pathway to pro-inflammatory and innate immune responses present an opportunity to understand the pathophysiology behind brain necrosis. An improved understanding of this pathway could provide therapeutic insights and opportunities for drug discovery. While SRS alone is well tolerated, several studies suggest that immunotherapy and chemotherapy may increase toxicity. fSRS and more stringent dosimetric guidelines may be appropriate in the setting of certain systemic therapies, particularly when delivered concurrently. The dependence on surgical biopsy to distinguish necrosis from recurrent tumor remains a major obstacle for patient management and provides a rationale for innovative approaches. Novel imaging modalities coupled with advancements in computer vision, radiomics, and deep learning represent promising opportunities. Peripheral blood cell-free DNA methylation data could reveal cell-specific insights into tumor biology and soon provide clinicians with a non-invasive alternative to surgical biopsy for diagnosis. These nascent approaches warrant further investigation. Awareness of the clinical challenges and opportunities associated with radiation necrosis in the setting of brain metastases is essential as clinicians are tasked with the delicate balance of optimizing tumor control and minimizing adverse effects in the modern era of targeted therapy and immunotherapy.

## Figures and Tables

**Figure 1 cancers-15-02432-f001:**
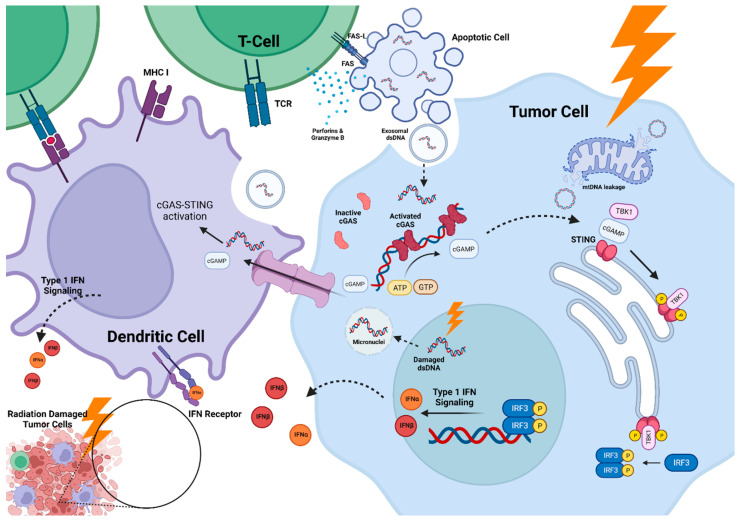
cGAS-STING Pathway: Radiation-induced DNA damage leads to accumulation of double-stranded DNA (dsDNA) in the cytosol of tumor, stromal, endothelial, and immune cells via (1) generation of micronuclei, (2) leakage of mitochondrial DNA (mtDNA), and (3) uptake of extracellular dsDNA from neighboring apoptotic cells via exosomes. Cytosolic cGAS undergoes reconfiguration and dimerization to form an activated state when bound to dsDNA. This allows the conversion of ATP and GTP to 2′,3′-cGAMP, which acts as a messenger molecule. Both cGAMP and dsDNA can act as paracrine signaling factors to activate cGAS-STING in neighboring cells, including dendritic cells (DCs). In the presence of cGAMP, STING located on the endoplasmic reticulum oligomerizes with TBK1 and undergoes phosphorylation. This then allows phosphorylation and activation of IRF3, which translocates to the nucleus to induce type 1 IFN signaling, which then targets DCs and other myeloid cells. By this mechanism, cGAS-STING mediates DC maturation, migration, and costimulatory molecule expression, including the expression of MHC 1 receptors. This leads to the priming of CD8+ cytotoxic T-cells, which then induce target cell death via granzymes, perforins, and activation of death receptor signaling. Thus, cGAS-STING serves as a crucial bridge between radiation-induced DNA damage and both innate and adaptive anti-tumor immune responses. Figure created with BioRender.com.

**Figure 2 cancers-15-02432-f002:**
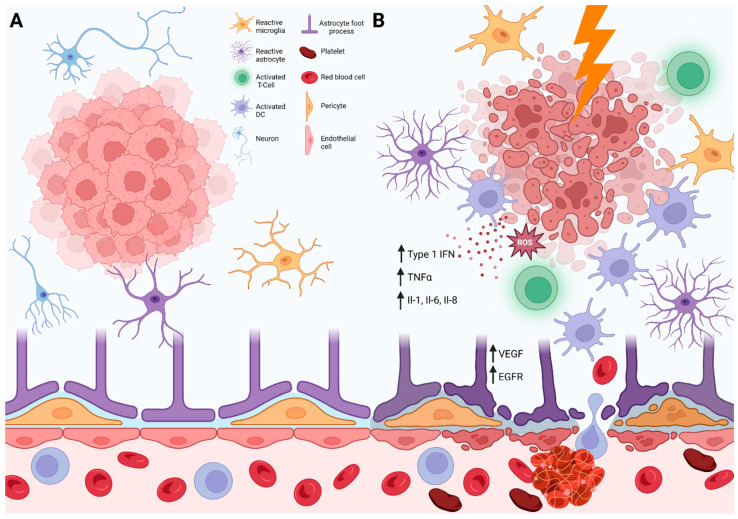
Radiation necrosis pathophysiology. (**A**) The tumor microenvironment of a brain metastasis prior to radiation includes a conglomerate of tumor cells surrounded by resident microglia, astrocytes, and neurons. Circulating immune cells and red blood cells are separated from the brain parenchyma by an intact blood-brain barrier made up of endothelial cells and pericytes. (**B**) The tumor microenvironment following radiation therapy is characterized by the upregulation of pro-inflammatory and innate immune responses. In the setting of necrotic tumor cells, reactive oxygen species (ROS), type 1 IFNs, TNFα, and interleukins are upregulated. Gliosis, with reactive astrocytes and microglia, is seen. Damage to pericytes, endothelial cells, and other resident cell populations leads to hypoxia-induced VEGF and EGFR expression. A leaky blood-brain barrier allows for migrations of macrophages, dendritic cells, and cytotoxic CD8+ T-cells into the tumor microenvironment. Hemorrhage and thrombosis are additional hallmarks of necrosis. Figure created with BioRender.com.

**Figure 3 cancers-15-02432-f003:**
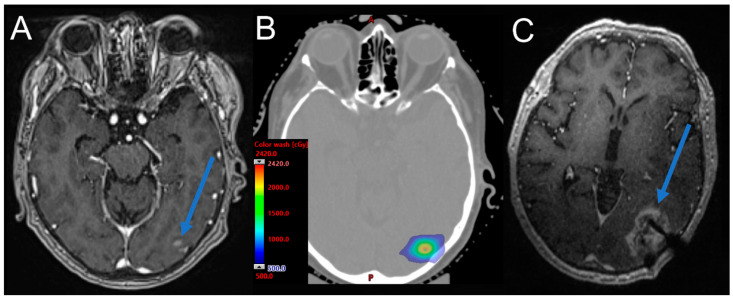
Representative case of radiation necrosis following concurrent CTLA4/PD1 inhibition with SRS. (**A**) MRI confirms new left occipital melanoma metastasis (blue arrow). (**B**) The SRS treatment plan delivered several days later, after the receipt of ipilimumab/nivolumab. A and P (in red), denote anterior and posterior, respectively. (**C**) MRI demonstrating localization of biopsy-confirmed necrosis at the site of the previous left occipital metastasis (blue arrow) 16 months after SRS treatment. Radiation necrosis occurred within the high dose treatment volume.

**Table 1 cancers-15-02432-t001:** Immunotherapy and Radiation Studies.

Immunotherapy Studies											
Author	Study Type	Sample Size *	Histology	Radiation	Systemic Therapy	Timing of Immunotherapy	Follow Up (Months)	Any Grade Necrosis Rate *	Biopsy Rate *	Symptomatic Necrosis Rate *	Necrosis Predictors
Colaco (2016)	Retrospective	180	Lung 39% Melanoma 31% Breast 15%	SRS	ICI only: 18% TT only: 11% CT only: 46% None: 2%	Not reported	11.7	39 (22%) ICI only: 12 (38%) TT only: 5 (25%) CT only: 14 (17%)	11 (28%)	18 (46%)	ICI (OR: 2.71)
Diao (2018)	Retrospective	91	Melanoma 100%	SRS + ICI vs. SRS	56% Ipilimumab	25% concurrent (≤4 wks)	7.4	SRS alone: 1 (3%) Concurrent: 2 (9%) Delayed: 2 (7%)	5 (100%)	5 (100%)	Not reported
Martin (2018)	Retrospective	480	NSCLC 61% Melanoma 30%	SRS/fSRS + ICI vs. SRS/fSRS	24% ICI (dual 0%)	Not reported	23.1 vs. 25.1	Not reported	Not reported	23 (20%) vs. 25 (7%)	IO (HR: 2.56) Melanoma (HR: 4.02)
Du Four (2018)	Retrospective	43	Melanoma 100%	RT + ICI (72% SRS, 19% WBRT)	100% Pembro (dual 0%)	SRS before Pembro (72%)	50	5 (12%*)*	3 (60%)	5 (100%)	Not reported
Kluger (2019)	Prospective	23	Melanoma 100%	Prior RT (SRS 52%, WBRT 22%)	100% Pembro	Not reported	24	7 (30%)	5 (71%)	Not reported	Not reported
Minniti (2019)	Retrospective	80	Melanoma 100%	SRS/fSRS + ICI	56% Ipilimumab 44% Nivolumab (dual 0%)	100% concurrent (≤1 wk)	15	28 (35%)	5 (18%)	12 (43%)	GTV (for symptomatic necrosis)
Kowalski (2020)	Retrospective	179	NSCLC 70% Melanoma 6%	SRS + ICI vs. SRS	20% ICI (dual 2%)	100% concurrent (≤3 months)	7.7 vs. 10.3	Not reported	None	1 (4%) vs. 10 (7%)	Tumor size ≤ 2 cm (HR: 0.24)
Goldberg (2020)	Prospective	42	NSCLC (100%)	21 with prior RT (SRS 38%, WBRT 19%)	100% Pembro	19% ≤3 months	8.3	3 (14%)	Not reported	1 (33%)	Not reported
Vaios (2022)	Retrospective	206	NSCLC 56% Melanoma 44%	SRS/fSRS + ICI vs. SRS/fSRS	75% ICI (dual 26%)	49% concurrent (≤4 wks)	15.3	Dual: 78 (26%) lesions Single: 87 (14%) No ICI: 39 (13%)	25 (12%) lesions	Dual: 20 (36%) Single: 17 (17%) SRS alone: 7 (14%)	Melanoma (HR: 2.41) fSRS (HR: 0.27) Concurrent ICI (HR: 6.47)
Lehrer (2023)	Retrospective	657	NSCLC 57% Melanoma 36%	SRS + ICI	100% ICI (dual 16%)	44% concurrent (≤4 wks)	13.4	66 (10%)	5 (8%)	45 (68%)	V12Gy (per RPA analysis)

* Units are patients unless otherwise specified.

**Table 2 cancers-15-02432-t002:** Targeted Therapy and Radiation Studies.

Targeted Therapy Studies											
Author	Study Type	Sample Size *	Histology	Radiation	Systemic Therapy	Timing of Targeted Therapy	Follow Up (Months)	Overall Necrosis Rate *	Biopsy Rate *	Symptomatic Necrosis Rate *	Necrosis Predictors
Narayana (2013)	Retrospective	12	Melanoma 100% (V600 mutant)	RT + TT (58% SRS, 25% WBRT)	100% vemurafenib	42% concurrent (during RT)	12.2	1 (8%)	None	1 (100%)	Not reported
Cho (2020)	Retrospective	379	NSCLC 100%	SRS +/− ICI/TT	ICI: 18% TT: 15% TT + ICI: 4%	Not reported	10.7	28 (7%)	None	Not reported	Not reported
Parsai (2020)	Retrospective	126	HER2+ Breast 100%	SRS + TT vs. SRS	37% lapatinib	19% concurrent (≤5 d)	17.1	1-year 1% vs. 6%	None	Not reported	Tumor volume
Popp (2020)	Prospective	124	NSCLC 52% Melanoma 14% Breast 19%	HA-WBRT + SIB vs. WBRT	ICI: 6% TT: 28%	73% concurrent (during or after RT)	8.5 vs. 6.3	HA-WBRT + SIB: 27 (7%) lesions	HA-WBRT + SIB: 2 (7%) lesions	HA-WBRT + SIB: 2 (3%)	Not reported
Park (2022)	Retrospective	46	HER2+ Breast 100%	SRS/fSRS + TT/Chemo	100% TT/chemo	59% concurrent (≤4 wks)	>12	28 (61%)	10 (36%)	Not reported	Multiple HER2-directed agents

* Units are patients unless otherwise specified.

**Table 3 cancers-15-02432-t003:** Chemotherapy and Radiation Studies.

Chemotherapy Studies												
Author	Study Type	Sample Size *	Histology	Radiation	Systemic Therapy	Chemo Agents	Timing of Chemotherapy	Follow Up (Months)	Overall Necrosis Rate *	Biopsy Rate *	Symptomatic Necrosis Rate *	Necrosis Predictors
Minniti (2011)	Retrospective	206	Lung 51% Melanoma 17% Breast 18%	SRS	76% Chemo	Not reported	100% before or after SRS	9.4	75 (24%) lesions	12 (16%) lesions	31 (41%) lesions	Tumor volume V10 Gy V12 Gy
Sneed (2015)	Retrospective	435	Lung 40% Melanoma 14% Breast 31%	SRS	59% of lesions (ICI: 2% Chemo: 38% TT: 9%)	Gemcitabine, capecitabine, vinca alkaloids, antifolates, taxanes, platinum agents, topoisomerase inhibitors	100% concurrent (≤1 month)	9.9	118 (5%) lesions	17 (14%) lesions	71 (60%) lesions	Prior SRS or WBRT (HR: 3.7) Target volume (HR: 1.1) Taxane (HR: 0.3) Capecitabine/5FU (HR: 2.6)
Kohutek (2015)	Retrospective	160	NSCLC 43% Melanoma 23% Breast 16%	SRS	44% Chemo	Not reported	100% concurrent (≤8 wks)	17.2	70 (26%) lesions	22 (31%) lesions	47 (67%) lesions	Tumor diameter (HR: 3.1)
Cagney (2018)	Retrospective	149	NSCLC 100%	SRS/fSRS + pemetrexed vs. SRS/fSRS + Other Chemo	100% Chemo	Pemetrexed (70.5%) or Other (26.8%); 91% received alternative platinum doublet	100% after SRS	24 vs. 20	24% vs. 10% at 1 year	None	Not reported	Pemetrexed (HR: 2.7)
Siddiqui (2019)	Retrospective	198	Lung 62% Melanoma 4% Breast 21%	SRS	ICI: 1% Chemo: 89% TT: 10%	Not reported	100% concurrent (≤3 months)	24	55 (8%) lesions	10 (18%) lesions	33 (60%) lesions	Tumor volume (HR: 1.1) Female gender (HR: 0.5)
Di Perri (2020)	Retrospective	294	Lung 56% Melanoma 4% Breast 17%	fSRS	ICI: 7% Chemo: 44% Other: 35%	Not reported	100% concurrent (≤3 months)	16.8	33 (9.2%) lesions	None	17 (51.5%) lesions	27 Gy/3 fx (HR: 3.07) 35 Gy/5 fx (HR: 4.22) ICI (HR: 2.69)

* Units are patients unless otherwise specified.

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
