# Peer review of "Novel Mechanisms and Future Opportunities for the Management of Radiation Necrosis in Patients Treated for Brain Metastases in the Era of Immunotherapy"

_cancers, 2023, doi:10.3390/cancers15092432_

Round 1

Reviewer 1 Report

The phenomenon of radiation necrosis is a type of tissue damage that can occur in the brain following stereotactic radiotherapy (SRS) for brain metastases. As more patients undergo this treatment and combine it with systemic therapy, the incidence of necrosis has increased. The authors explained how the cGAS-STING pathway plays a key role in linking radiation-induced DNA damage to inflammation and immune response, and may contribute to the development of necrosis. There are potential targets for therapeutic development and highlights advancements in management strategies, including dosimetric strategies, imaging, biomarkers, and artificial intelligence. The authors also discussed the biological mechanism linking radiation-induced DNA damage to pro-inflammatory effects and innate immunity, which could play a key role in the pathogenesis of necrosis. However, there is little new insight and little novelty throughout. A review article may propose new hypotheses or ideas that have not been previously considered. And then new ideas can be tested in future research, potentially leading to breakthroughs.

Author Response

Response to Review:

Novel Mechanisms and Future Opportunities for the Management of Radiation Necrosis in Patients Treated for Brain Metastases in the Era of Immunotherapy” (2314010)

We are thankful for the Editor’s time and consideration. We also thank the Reviewers for their insightful and overall positive comments. We believe that we were able to address all points raised, and that the manuscript has significantly improved as a consequence of these changes. Please find our point-by-point response below.

Reviewer #1:

  • The phenomenon of radiation necrosis is a type of tissue damage that can occur in the brain following stereotactic radiotherapy (SRS) for brain metastases. As more patients undergo this treatment and combine it with systemic therapy, the incidence of necrosis has increased. The authors explained how the cGAS-STING pathway plays a key role in linking radiation-induced DNA damage to inflammation and immune response, and may contribute to the development of necrosis. There are potential targets for therapeutic development and highlights advancements in management strategies, including dosimetric strategies, imaging, biomarkers, and artificial intelligence. The authors also discussed the biological mechanism linking radiation-induced DNA damage to pro-inflammatory effects and innate immunity, which could play a key role in the pathogenesis of necrosis. However, there is little new insight and little novelty throughout. A review article may propose new hypotheses or ideas that have not been previously considered. And then new ideas can be tested in future research, potentially leading to breakthroughs.

Response: We acknowledge Reviewer 1’s assessment and agree with the need to highlight new insights and novelty, in addition to proposing new hypotheses and ideas for future research, in a review. We edited the manuscript throughout to emphasize novel concepts, hypotheses, and gaps in the literature. In the subsequent topics, we specify these updates:

cGAS-STING: This pathway has only recently been implicated as a potential mechanism involved in radiation necrosis. We emphasize the novelty of this line of research in lines 92-95 and 164-168. We now specifically discuss the importance of preclinical models (lines 183-192) of radiation necrosis to elucidate the importance of various hypothesized mechanisms (e.g., cGAS-STING, endothelial injury, gliosis, tumor immunogenicity). The lack of an animal model for necrosis is a barrier for development of therapeutic strategies.

Dosimetric strategies: In lines 500-503, we recommend that investigators consider revising dosimetric guidelines for high risk patients (e.g., reirradiation, concurrent immunotherapy). The lack of a personalized dosimetric strategy to achieve tumor control without unwanted toxicity by considering prior radiation history and systemic therapy remains an unmet need and an avenue for research. 

Imaging Techniques: In lines 529-535, we expand our recommendations for imaging-based research. Specifically, we underscore the need for standardized imaging criteria for new modalities to enable better interpretability between institutions. We now highlight the Deauville Score as a template for imaging criteria that are not limited by institutional differences in image acquisition protocols and interpretation.

Artificial Intelligence: In lines 563-568, we hypothesize that integration of biological data will outperform pure radiomic approaches. We additionally highlight the need for prospective data and training-sets that use only biopsy proven cases.

Circulating biomarkers: In lines 570-571, 581-583, 595-596, 604-609, we outline several hypotheses and novel lines of research. We particularly emphasize the hypothesis that methylation patterns derived from cell-free DNA, which are cell-of-origin specific, may provide insight into tumor biology. We also discuss the use of deconvolution methods to infer the activation state of circulating immune cells, which may offer an immune signature for necrosis. The need for multi-institutional collaboration, likely via U01 grants, is additionally underscored.

Reviewer 2 Report

The ms by Vaios et al. presents very comprehensively current radiotherapeutic approaches in brain metastases, as well as mechanisms of one of the main RTX effects - radionecrosis.

The ms is well organized, adequately referenced, and contains scoping figures and summarizing tables.

I would have just a few minor points for correction:

1. line 62 - please also mention CyberKnife technology

2.

Table 1 - it is not necessary to repeat "patients" in each line (columns 3 and 11) - please transfer to column heading; column 10 - the same if lesions = patients
Table 2 - as above (columns 3, 9, 10)
Table 3 - as above (column 3, and "lesions" in 8, 9, 10)
A1-A3 - same concept

3. There are a few grammatical/punctuation/spelling/stylistic mistakes - need one thorough reading.

Author Response

Response to Review:

Novel Mechanisms and Future Opportunities for the Management of Radiation Necrosis in Patients Treated for Brain Metastases in the Era of Immunotherapy” (2314010)

We are thankful for the Editor’s time and consideration. We also thank the Reviewers for their insightful and overall positive comments. We believe that we were able to address all points raised, and that the manuscript has significantly improved as a consequence of these changes. Please find our point-by-point response below.

Reviewer #2:

  • Line 62 - please also mention CyberKnife technology.

Response:  We agree with the Reviewer and have updated this accordingly.

  • Tables 1-A3: it is not necessary to repeat "patients" in each line (columns 3 and 11) - please transfer to column heading; column 10 - the same if lesions = patients.

Response: We agree with the Reviewer and have updated all tables accordingly. An asterisk is now included denoting the units of measurement.

  • There are a few grammatical/punctuation/spelling/stylistic mistakes - need one thorough reading.

Response: The manuscript has been reviewed for clarity, spelling, grammar, and style.

Reviewer 3 Report

Authors explained the study well

1. I believe it’s important to explore types of brain metastases patients getting. ( types of brain cancer and introduction ) it will be easy for the readers to understand 

2. Cancer stem cells are the major reason for disease reoccurrence or metastasis conditions. They are resistant to chemotherapy or radiotherapy. Please include some information regarding CSC.
